# Fusion Object Detection and Action Recognition to Predict Violent Action

**DOI:** 10.3390/s23125610

**Published:** 2023-06-15

**Authors:** Nelson R. P. Rodrigues, Nuno M. C. da Costa, César Melo, Ali Abbasi, Jaime C. Fonseca, Paulo Cardoso, João Borges

**Affiliations:** 1Engineering School, University of Minho, 4800-058 Guimarães, Portugal; 2Algoritmi Center, University of Minho, 4800-058 Guimarães, Portugal; 3Polytechnic Institute of Cávado and Ave, 4750-810 Barcelos, Portugal; 42Ai—School of Technology, Polytechnic Institute of Cávado and Ave, 4750-810 Barcelos, Portugal

**Keywords:** machine learning, visual intelligence, object detection, image processing, action recognition, autonomous vehicles

## Abstract

In the context of Shared Autonomous Vehicles, the need to monitor the environment inside the car will be crucial. This article focuses on the application of deep learning algorithms to present a fusion monitoring solution which was three different algorithms: a violent action detection system, which recognizes violent behaviors between passengers, a violent object detection system, and a lost items detection system. Public datasets were used for object detection algorithms (COCO and TAO) to train state-of-the-art algorithms such as YOLOv5. For violent action detection, the MoLa InCar dataset was used to train on state-of-the-art algorithms such as I3D, R(2+1)D, SlowFast, TSN, and TSM. Finally, an embedded automotive solution was used to demonstrate that both methods are running in real-time.

## 1. Introduction

Mobility could be defined as the potential for movement and the ability to get travel from one place to another using one or more modes of transport to meet daily needs. The transportation mode can be called a vehicle. In recent years, a new technological iteration in the evolution of vehicles has been integrated into mobility, using autonomous driving. Autonomous vehicles have different levels of automation (Level 0 to Level 5), and Level 5 is commonly referred to as autonomous, self-driving vehicles. The term “autonomous” is inconsistently used in the literature, but some state legislation refers to highly automated driving systems, that is, at or above Level 3 [1,2]. Shared mobility refers to the shared use of a vehicle for the carrying out of a trip, such as car sharing, bike sharing, scooter sharing, on-demand ride services, and ride sharing [3]. Although there exist several vehicles available for our shared mobility, the car is still by far the most frequently used transportation mode that humans rely on, and it will be the topic for this article. Within this topic, the concept of shared autonomous vehicles (SAVs) is expected to gain a huge preponderance, as it will allow for passenger transportation in an easy, fast, and economical way. The absence of a human driver in SAVs has raised concerns about passenger safety, making passenger monitoring a critical aspect to ensure the well-being of both passengers and the vehicle. Detecting incidents of passenger–passenger violence is particularly relevant in this context, which requires careful consideration of the SAV concept to guarantee the exclusivity of passengers as human agents and to ensure their safety is not compromised. On the other hand, it will make it possible for any passenger action that jeopardizes the vehicle’s integrity to be identified (passenger–vehicle interaction), safeguarding the interests of the company providing the service. To implement this monitoring system, it is essential to develop an algorithm method to recognize human actions inside the car. This type of method comprises two subclasses: traditional methods and deep learning methods. Traditional methods require a manually implemented feature extraction module, and deep learning methods automatically learn to extract the features that best characterize the image. Although deep learning-based methods achieve results with state-of-the-art accuracy compared to traditional methods, they also require a lot of data to process the algorithms, which implies the need to use or to create datasets to feed them.

Since deep learning-based monitoring algorithms are required to run in real-time, they must be energy efficient so as not to jeopardize the autonomy of the vehicle. Going into the aspect of implementing deep learning algorithms in embedded systems, it is possible to verify that different approaches bring benefits and constraints. Systems such as field-programmable gate arrays (FPGAs) allow for the implementation of a logic circuit that is dedicated and specific to the specificity of the developed algorithm. However, they lead to greater development complexity and the risk of not obtaining results that are equal or superior to other systems that offer the facilitated of parallel execution. Another advantage of these systems is the ability to obtain a reduction in consumption compared to other systems. As an alternative, some systems focus on a generic architecture and are supported by graphic processing units (CUDA cores), providing easy access to the parallelization of instructions, and thus achieving rapid deployment at the expense of greater energy consumption. This solution is typically used to train deep learning-based methods. Both systems mentioned are possible to find in the industry for the most varied needs; however, they end up lacking ASIL or SAE. It is worth noting that it may be the focus of automakers to find a solution that can safeguard the safety of passengers by providing automotive solutions that enable monitoring systems to perform real-time inference, and at the same time, ensure that the energy consumption of this solution is not too high. We propose an embedding object detection and violent action detection into one pipeline to achieve the best possible performance in human violent action recognition. For object detection data, we introduced the procedure of collecting and restructuring public datasets, namely, COCO [4] and TAO [5], which we fused to create the MoLa InCar dataset [6], an aggressive action and object custom dataset. In object detection, we tested different versions of the YOLO detector and selected one of the active-development SOTAs, YOLOv5 [7]. From these studies, we were able to define the best object detection method for lost items and aggressive objects, which can be easily inserted into complex video processing pipelines for existing projects. For action recognition, we benchmark our dataset [6] end-to-end features (RGB, event frames, and optical flow,) using SOTA 2D and 3D CNN models. To conclude, we present our final solution, running all the algorithms in real-time on an embedded system. The architecture of the proposed solution is illustrated in Figure 1. The remainder of the paper is organized as follows. Section 2 presents the state-of-the-art, presented with respect to the automotive industry, focusing on autonomous vehicle developments. The deep learning-based algorithmic solution is used for action recognition and object detection. The chapter that concludes that it mentions the deployment solutions to embed the algorithms for in-car scenarios. Section 3 described the preparation of the dataset with the subtasks performed in the development of the dataset preparation to be fed into the algorithms. The algorithm analysis is presented in Section 4, which described the experiments performed for the violent action detection, as well as for the object detection algorithm, namely the aggressive objects. Section 5 presented the results of the algorithms that allow for the selection of the best algorithms with the highest precision and the lowest computational requirements, and also described the pipeline used for the embedded solution, which will process both algorithms. Finally, Section 6 provides general conclusions and a discussion.

## 2. Related Work

The development and introduction of fully autonomous vehicles is seen as the future of the automotive industry. With the implementation of this new paradigm, a revolution is foreseen in the way that transportation are used today. Actions by autonomous vehicle manufacturers and partners suggest that these vehicles will initially be deployed in shared mobility services. For example, BMW, Ford, Volkswagen, and Hyundai have all partnered with various companies to develop autonomous vehicles for ride-sharing and on-demand services, with production being planned in 2021 [8,9,10]. Daimler has partnered with Uber to allow for the introduction of autonomous vehicles in Uber’s ride-sharing network [6]. Toyota has also partnered with Uber with the same goal [11]. Waymo has already started commercial autonomous ride-sharing services in Tempe, Mesa, and Chandler [12].

There are several developments regarding to the monitoring of human motion using sensor fusion, such as the work presented by Abbasi et al. [13] although it is does not focuses for the in-car monitoring, demonstrating the potential of using sensor fusion and deep learning techniques for multi-human tracking by using NVS sensors. In [14], Melo et al. use a similar approach of sensor fusion, in this case, by combining RGB and thermal cameras with deep learning techniques for monitoring the presence of masks on people in public spaces in the context of COVID-19 and for measuring the body temperature.

Regarding to the works on the in-car environment, the work proposed by Torres et al. [15] can detect the human body pose inside the vehicle by using a time-of-flight sensor which that could provide more light immunity compared to RGB sensors. In [16], Dixe et al. use a similar approach to detect multi-person human body detection by using depth images generated synthectically with the knowledge of the work developed by Borges et al. in [17,18]. Although, in [19], Dixe et al. follow a different approach by using generative adversarial networks (GANs) for automatically generating artificial images of vehicle interiors to support the developed algorithms for the creation of monitoring systems.

In [20], Dixe et al. presented a monitoring system capable of estimating the state of the car, namely, the presence of damage, dirt, and stains using semantic segmentation. A similar work was developed by Faria et al. in [21] by classifying the state of the vehicle interior with the use of state-of-the-art classifiers using RGB images.

Regarding action recognition, there is a lot of work being developed in this field. In [22], Carreira et al. introduced a deep learning model and kinetics datasets, which is a large-scale video action recognition dataset. The model is called two-stream inflatable 3D ConvNet (I3D), and it is an extension of the popular two-stream architecture that uses spatial and temporal information. Their main contributions are the proposed I3D model, which is a 3D CNN that inflates 2D filters to 3D, and that uses spatial and temporal information. We can not only reuse the 2D models’ architecture (e.g., ResNet and Inception), but we can also bootstrap the model weights from the 2D pretrained models. Following the same path of spatio-temporal 3D ConvNets is the model R(2+1)D.

Tran et al. [23] present a deeper analysis of the use of spatio-temporal convolutions for action recognition in videos. Spatio-temporal convolutions are a type of convolutional layer that can model spatial and temporal dependencies in the input data. They are commonly used in video action recognition models and are effective for this task. Their main contributions are a more detailed analysis of the properties of spatio-temporal convolutions and their impacts on the performance of action recognition models. The authors study different variations of spatio-temporal convolutions, such as 2D convolutions, 3D convolutions, and separable spatio-temporal convolutions. They also evaluated the impacts of different factors such as kernel size, dilation rate, and the number of layers, on the performance of the models. They trained and tested their algorithm with Sports1M [24], Kinetics [25,26,27], UCF101 [28,29], and the HMDB51 [30] datasets, reporting an accuracy of 73.3%, 72%, 96.8%, and 74.5%, respectively. SlowFast is another recent implementation of the Resnet3D backbone, and it was presented by Feichtenhofer et al. [31] for video action recognition. SlowFast Networks are a type of two-stream architecture that uses both a slow pathway, which processes the video at a lower frame rate, and a fast pathway, which processes the video at a higher frame rate. The technique is partially inspired by the retinal ganglion in primates, in which 80% of the cells (P-cells) operate at a low temporal frequency and recognize fine details, and 20% of the cells (M-cells) operate at a high temporal frequency and are responsive to swift changes. Similarly, in SlowFast, the compute cost of the Slow pathway is 4x larger than that of the Fast pathway. Their main contributions are the proposed SlowFast Networks, which can be trained end-to-end to learn both spatial and temporal representations of the video. The slow pathway captures long-term temporal information, while the fast pathway captures fine-grained temporal information. The authors also propose a new fusion strategy that combines the outputs of the two pathways to make the final prediction. The proposed method was evaluated on several action recognition datasets, and it outperforms state-of-the-art methods that use only a single pathway or simple fusion methods. In [32], the authors present a method for action recognition in videos called Temporal Segment Networks (TSNs). TSNs is a deep learning method that addresses the problem of recognizing actions in videos with variable length and temporal structure. It combines a sparse temporal sampling strategy and video-level supervision to enable efficient and effective learning, using the whole action video. Their main contributions are the TSN architecture, which is composed of multiple branches that process different segments of the input video, and a fusion module that combines the features of all branches to make the final prediction. The authors also propose several good practices for training TSNs, such as using multiple segments per video, data augmentation, and a consensus loss function. In [33], Lin et al. proposed a new method called Temporal Shift Module (TSM) for video action recognition. TSM is a technique that enables the network to efficiently process videos of different lengths by shifting the frames of the video in the temporal dimension. Their main contributions are the proposed TSM module, which can be added to existing CNN architectures and enables the network to efficiently process videos of different lengths by shifting the frames of the video in the temporal dimension. Video understanding faces the challenge of achieving a high accuracy at a low computational cost. While 2D CNNs are computationally cheap, they fail to capture temporal relationships. Meanwhile, 3D CNNs are computationally intensive but achieve a high performance. The Temporal Shift Module (TSM) offers a solution that combines high efficiency with performance by enabling the exchange of information among adjacent frames, achieving 3D CNN performance while retaining 2D CNN complexity. TSM can be easily integrated into 2D CNNs without adding any computational or parameter costs. TSM is also adaptable to online settings, enabling real-time and low-latency recognition and detection of video objects. The authors also propose a new architecture that uses the TSM module to process the input video and to show that it can improve the performance of the model while reducing computational costs.

Object detection is a well-studied area with diverse applications, including mask detection. The R-CNN family of algorithms [34,35] identifies regions of interest and uses CNN to detect objects within those regions. More recently, YOLO [36], developed by Redmon et al., introduces a novel object detection system called YOLO (You Only Look Once), which is based on a neural network that takes an entire image as input and outputs a set of bounding boxes and class probabilities for all objects in the image. This is in contrast to traditional object detection systems that use multiple stages to identify objects, such as region proposal generation, feature extraction, and object classification. The YOLO object detection family presented as YOLOv2 [37], YOLOv3 [38], YOLOv4 [39], and YOLOv5 [7], provides a more accurate and faster method compared to the R-CNN family.

In addition to the development of a method for action recognition in a car environment, it is necessary to consider its incorporation into it. To do this, it is necessary to select and to develop an embedded system that allows for the implementation of the algorithm and that is suitable for the automotive context. Embedded computing systems that are selected to operate in a vehicle should have certain factors as a selection focus, these being the cost and the ratio between the operations per second and the energy consumed (FLOPS/Watt). Even considering the selection based on the mentioned factors, it is relevant to focus on the development, taking into account the future trends of the automotive industry. With the prospects of achieving full autonomous driving levels, car control systems are moving towards a centralized topology, where all the intelligence of the car is composed of a single processing unit. To meet this general requirement of the automotive industry, NVIDIA provides its products for the centralized development of all vehicle intelligence. All the factors mentioned above, such as cost, computing, energy expenditure, ASIL, and SAE are available for the different needs of autonomous driving. Right now, NVIDIA is a tier 1 company that serves OEM customers who make autonomous cars. Implementations and direct comparisons between standard and embedded CUDA systems show that it is possible to achieve the same performance with a 50% reduction in power consumption [40,41].

## 3. Dataset Preparation

This topic described the subtasks performed in the development of the dataset preparation to be fed into the algorithms. In Figure 2 summarizes the entire development pipeline of this article.

### 3.1. MOLAnnotate Framework

For the faster implementation of algorithms, we used the MoLAnnotate Toolkit https://github.com/eng-motionlab/molannotate (accessed on 14 December 2022), an open-source framework for dataset annotation software to merge, fuse, split, check, and export datasets (videos or/and images) to different algorithms.

#### 3.1.1. Unified Format

The most significant part of the toolkit is in developing a data format that is both comprehensive and easy to manipulate. As such, we base our JSON format on the COCO format, as can be seen in Figure 3. The simple definition of JSON is a collection of name-value pairs in object format (e.g., name: value) [4].

For the metadata, we defined the name ‘info’ to provide a description of the file, and the name ‘licenses’ to save a list of all the licenses for the images and videos. The following names specifically belong to the annotation of the dataset. An example of the structure of the list object is shown in Figure 3 ‘datasets’, where two datasets are listed using dictionaries with ‘name’ and the correspondent ‘id’. All the other names follow the same structure but with different purposes. The categories/classes of the labeling of the dataset are listed in ‘categories’; each category has at least the following dictionary structure ‘id’: int, ‘name’: str, ‘dataset’: int—note that int, str, and [] are all variable types of Python. Similarly, all image information is stored in a list of dictionaries in ‘images’, videos in ‘videos’, tracking information in ‘tracks’, annotations for each image with bounding boxes for each category in ‘annotations’, and annotations for videos, such as the original start frame of the labeling and the end frame, in ‘video_annotations’.

#### 3.1.2. Annotation Pipeline

The annotation pipeline consists of 5 types of algorithms: merge, fusion, split, check, and export (Table 1).

### 3.2. Violent Action

For the violent action, we used the MoLa InCar dataset [6]. This dataset was recorded with RGB, Depth, Thermal, and Event-based data (see Figure 4). Although it was recorded in a laboratory environment, the actions were performed on a car testbed (see Figure 5). The resulting dataset contains 6400 video samples and more than 3 million frames, collected from 16 distinct subjects (different ages, genders, and heights). We also provided information related to the clothing color, material, and skin tone, according to the Fitzpatrick scale [42] of each subject for each recording. The dataset contains 58 action classes, including violent and neutral (i.e., nonviolent) activities. Although our labeling is the primary binary for the moment (violent and nonviolent), it fits the purpose of detecting violence inside the vehicle.

To train violent action models, we labeled the frames ‘VIOLENT’ and ‘NONVIOLENT’, and used RGB, Thermal, NVS, and optical flow frames for end-to-end training. Using MOLAnnotate, we started by ‘dataset2json.ipynb’ to convert the data and labels of the dataset into the MOLA annotation format. This script imports the categories, videos, and images, and creates the respective annotations.

The resultant ‘INCAR.json’ (see Figure 6) had 760 original videos annotated, 1551 video annotations (760 videos split into 745 ‘VIOLENT’ and 806 ‘NONVIOLENT’ labeled clips), and 443,433 images (annotated only with one of the two classes, ‘VIOLENT’ or ‘NONVIOLENT’).

Then, we used the ‘json2mmaction2.ipynb’ to export our dataset to the format of the mmaction2 algorithms methods (Figure 6). This is a framework that can be used to train end-to-end action recognition models. In this case, ‘json2mmaction2.ipynb’ exports the dataset, generates a file list for all videos, and splits the file list into train lists (50%, 4 pairs, subjects P1–P8), validation (25%, 2 pairs, subjects P9–P12), and test (25%, 2 pairs, subjects P13–P16), as can be seen in Figure 6. Additionally, this all-in-one approach with the split function inside of the export script is one way of exporting, but in the next section, we give a different example of splitting the JSON into smaller training, validation, and testing JSONs, and then exporting.

### 3.3. Aggressive Objects

Another example of the requirements of training algorithms for detecting lost items and aggressive objects such as knives, weapons, and bats, versus non-aggressive personal objects such as a book, phone, or bag, we selected public datasets such as the COCO [4] and TAO [5] public datasets. COCO was assembled to tackle object recognition in the context of broader scene understanding. In this example pipeline, we used COCO 2017 annotations containing 80 object classes, with a total of 123,287 images and 886,284 instances labeled.

First, the public JSONs are merged into the MOLA format. Then, they are split by via annotation into train, validation (val), and test. Then, the 373 classes are fused into two different classes: aggressive and nonaggressive. TAO is a dataset for Tracking Any Object, containing 2907 high-resolution videos from other datasets (Charades, LaSOT, ArgoVerse, AVA, YFCC100M, BDD-100K, and HACS), captured in diverse environments, which are half a minute long on average and have tracks labeled for 833 object categories. The TAO dataset follows an annotation format similar to that of COCO [12]. In our pipeline, we imported 363 TAO object classes with annotations, with a total of 54,649 images and 167,751 instances labeled. We first prepare the entire dataset using the open source annotation pipeline developed by merging the datasets using ‘mergedatasets.ipynb’ (see Figure 7). After this, we start by fixing the classes with ‘fixclasses.ipynb’ (like, for example, removing duplicate categories), then cleaning the classes (“cleanclasses.ipynb”) with missing annotations and missing images, and cleaning the images (“cleanimages.ipynb”) that could be missing from the original directory of the datasets. The merged JSON ’cocotaolbo.json’ had 1488 videos, 194,943 images, 8123 tracks, and 1,348,451 annotations. Unlike the previous section example, we illustrated splitting the JSON into training, validation, and testing JSONs before exporting. As such, ‘cocotaolbo.json’ is split using annotations (“splitbyannotations.ipynb”, see Figure 7) that are balanced per category in 70% training annotations, 20% validation, and 10% for testing. Finally, the JSON is exported to YOLOv5 in this example. As such, for object detection, our requirements were a Multi-class Multi-object Detector (MCMOD). As such, we choose the You Only Look Once (YOLO) detector [36]. After testing different versions of YOLO, we selected the most recent state-of-the-art active development and stable Python framework, the YOLOv5 from Ultralytics [7]. Therefore, as an example, we export ‘cocotaolbo.json’ to train aggressive custom data.

After fusion, we use the script ‘json2yolo.ipynb’ to first export the dataset to the YOLOv5 structure, and then we generate the ‘filelist’ with the paths of the images. Customizing the dataset to feed YOLOv5 with aggressive data training, we can perform various experiments with different fusions of aggressive classes using ‘mixclasses.ipynb’. Additionally, note that fusion can be done before or after the split of JSONs. In this example pipeline, we fused aggressive classes (such as knife, fork, bat, bow, gun, weapon, and rifle) in one class, named ‘aggressive’, and the rest of the classes were fused in a ‘nonaggressive’ class (such as a person, car or book). Therefore, because we do not remove any class, the same number of annotations remains (as can be seen in Figure 8). These fusions are made in the script ‘mixclasses.ipynb’, resourced to an Excel report, where the user can select how to mix the classes. Then, we used ‘json2yolo.ipynb’ to export our dataset to the format of the YOLO algorithm [18], as shown in Figure 8. In this case, “json2yolo.ipynb” exports the dataset (images and labels) and generates a file list for each train, validation, and test json. This is a different approach from the previous section.

## 4. Algorithmic Analysis

The objective of this section is to define experiments for the recognition of violent actions and the detection of objects. All these tests were performed on a server with an Intel(R) Xeon(R) Gold 6140 CPU 2.30 GHz processor, 128 GB RAM, and NVIDIA Tesla V100-PCIE-16 GB computing GPU.

### 4.1. Methods

#### 4.1.1. Violent Action

The dataset customization was done during recording and labeling. Also, we were only able to label two classes, “VIOLENT” and “NONVIOLENT”. As such, no further customization was needed. In our experiments, we evaluate five state-of-the-art action recognition methods on the MoLa InCar dataset [6], the I3D [22], R(2+1)D [23], the Slow-Fast [31], TSN [32], and TSM [33]. We started by training our dataset in 3D convolutional networks. This is a simple, yet effective approach for spatio-temporal feature learning using deep 3-dimensional convolutional networks (3D ConvNets) trained on a large-scale supervised video dataset. 3D ConvNets are more suitable for spatio-temporal feature learning compared to 2D ConvNets.

E1—I3D: for our first experiment (E1), we started with the deep learning model introduced by Carreira et al. [22], the I3D. We trained the raw frames of each video using a sampling frame pipeline of clip length (Frames of each sampled output clip) × frame interval (Temporal interval of adjacent sampled frames) × num clips (Number of clips to be sampled) of 32 × 2 × 1. Also, this method was trained with the Resnet3d backbone without pretraining, and with a learning rate of 0.0025 for 100 epochs.

E2—R(2+1)D: following the same path of the spatio-temporal 3D ConvNets, in E2, we selected a more recent model R(2+1)D developed by Tran et al. [23]. This method was trained using a sampling frames pipeline of 32 × 2 × 1, the ResNet2Plus1d custom backbone, and no pre-trained weights, with a learning rate of 0.01875 for 100 epochs.

E3—SlowFast: for our third experiment (E3), we selected SlowFast with the Resnet3D backbone, which was presented by Feichtenhofer et al. [31]. This method was trained with the Resnet3dSlowFast backbone, with no pre-trained weights, and with a learning rate of 0.0125 for 100 epochs.

E4—TSN: concerning our method of labeling, we also decided to experiment with temporal ConvNets instead of spatio-temporal, such as the Temporal Segment Network (TSN); for this particular reason, we chose the work developed in [32] for our fourth experiment (E4). This method was trained with the Resnet50 backbone, without pre-trained weights, and with a learning rate of 0.0009375 for 100 epochs.

E5—TSM: with the good results obtained from TSN, we trained in E5 a faster state-of-the-art temporal action recognition model, the Temporal Shift Model (TSM) presented by Lin et al. in [33]. This method was trained with the ResnetTSM backbone, with no pre-trained weights, and with a learning rate of 0.001875 for 100 epochs.

#### 4.1.2. Aggressive Objects

For object detection, our requirements were a Multi-class Multi-object Detector (MCMOD). As such, we choose the You Only Look Once (YOLO) detector [36]. The YOLO system can perform object detection in real-time due to its single-stage architecture. The network is made up of convolutional layers and ends in a few fully connected layers that predict the bounding boxes and class probabilities. Using a single network, YOLO can achieve a high degree of accuracy while maintaining real-time performance, making it suitable for applications such as self-driving cars, robotics, and real-time surveillance. One of the advantages of the YOLO system is that it can learn from contextual information in the image, such as spatial relationships between objects. This allows the system to make more accurate predictions, and reduces false positives. Additionally, the YOLO system can be trained in a large number of object classes, making it highly adaptable to a wide range of applications. After testing different versions of YOLO, we selected a SOTA model, under active development, with a stable Python framework, the YOLOv5 from Ultralytics [7]. We then adapted the framework to our pipeline for training lost items and aggressive custom data. Customizing the dataset to feed YOLOv5 with aggressive data training, we performed various experiments with different fusions of aggressive classes.

#### 4.1.3. Embedded System

All models were implemented in PyTorch in the first iteration, exported as a weight file (weights.wts) or converted to Open Neural Network Exchange (ONNX), and then used TensorRT was used to load weights, define the network, and perform inference. TensorRT supports quantized floating points, which compresses and rounds floating-point numbers to 8-bit integers. This dramatically improves arithmetic throughput while lowering storage and memory bandwidth needs. To use the Int8 quantization it is necessary to prepare the calibration images, which are the ones used for the training; in our case, we selected 1000 images. Figure 9 shows the flow chart of how the algorithm for the embedded system was implemented to process the 3 methods (2 object detectors and 1 violent detector). First, we use the lost items objects detector, which has the person class and can detect if there is a person inside the vehicle; if there is none, we start to infer on any other object inside the vehicle. If there is at least one person inside the vehicle, the lost items object detector stops inferring and starts inferring the aggressive objects. Also, if there is more than one person inside the vehicle, the violent detection method starts to check if there is any violence between the passengers.

## 5. Results

This section can be divided into subtopics. It should provide a concise and precise description of the experimental results, their interpretation, and the experimental conclusions that can be drawn.

### 5.1. Experimental Evaluation

#### 5.1.1. Violent Action

The results of our experiments with the above-mentioned methods are reported in Table 2. The methods were trained with the same data with resized samples at 256p. Accuracy is the number of correct predictions divided by the total number of predictions (sum of the diagonal of confusion matrix, divided by the total number of predictions); Top-1 accuracy represents this accuracy, as it uses the highest probability prediction to match the expected answer (the sum of the diagonal of confusion matrix divided by the total number of predictions); Mean class accuracy, is calculated as the mean of the number of correct class predictions divided by the total number of class predictions (the mean of each confusion matrix column accuracy).

From the results, we were able to conclude that temporal action recognition worked better with our dataset labels context. Also, using TSM has the advantage of low latency (a faster inference time) for real-time applications. As such, we used TSM in the final embedded system.

In Table 3 presents the results for the detection of violent actions using different features for the TSN and TSM methods.

In conclusion, for a binary context, RGB is not needed to have good results. We can use optical flow and NVS to get similar results, and the training is faster. As such, violent action information is conserved through the different features. Additionally, we can see that temporal 2D CNNs have more performance than spatio-temporal 3D CNNs in this context of the binary categories of violent versus nonviolent. Due to the use of RGB frames for the aggressive objects and lost items algorithms, we maintain the use of the RGB feature for the violent action detection, otherwise, it was required two streams of process information from different sensors were required, which could lead to the increasing of the inference time for the recognition.

#### 5.1.2. Aggressive Objects

We started our experiment, E1, by feeding the same 76 classes to the YOLOv5s (model small, version 5.0) from scratch with default hyperparameters, with the results as seen in Table 4. Then, as the results were not significant, we decided to experiment (E2) using only 2 classes, the same aggressive class and the remaining 75 classes grouped in the nonaggressive class. The results were even worse, so we removed the person in the class that could lead to better results. In E3, class 1 (‘person’) has more than five times the annotations, which means that more or less, all images have one or more persons in them (see Figure 10).

Then, we checked in experiment E4 for if testing only the class aggressive (overfitting) would show the problem. As can be seen in Table 4, the results are only marginally better. As such, we demonstrate that mixing aggressive objects such as knives, weapons, bats, hammers, bows, and others is not a good policy, as they have very different characteristics. Afterward, we decided to test the separation of the best 20 aggressive classes using the best pre-trained default model YOLOv5x (E5), and this type of separation provided better results (as seen in the next section, Table 4, the mean of all the classes).

From these 20 classes, we selected those with more labels, bat (3000 labels) and knife (7000 labels), but we also decided to include weapons (500 labels) due to our case study (see Figure 11).

As such, in E9 (E6), from using the best YOLOv5x model (extra large) and testing it on 1337 images, we obtained the following results in Table 5.

This can also be inspected graphically in the confusion matrix (Figure 12), but the weapon class must be observed with some care, as we trained with only 500 labels. In fact, during real-time tests, the weapon inference was very low.

#### 5.1.3. Lost Item Objects

From the know-how obtained from the experiments of the previous aggressive pipeline, we were able to develop the lost items pipeline. In this pipeline, we selected 12 common object classes plus the 3 aggressive classes (knife, bat, and weapon) for a better separation of aggressive and non-aggressive: [‘person’, ‘knife’, ‘weapon’, ‘bat’, ‘bag’, ‘book’, ‘phone’, ‘laptop’, ‘mouse’, ‘keyboard’, ‘bottle’, ‘cable’, ‘banana’, ‘apple’, ‘orange’]. In Figure 13, we show the distribution of the labels.

In Table 6, you can see the results for each class using the second-best model YOLOv5l (large) model pre-trained, which is faster for real-time implementations.

The person class has the most labels; as such, without overfitting, such as the class weapon, it provides the best results, as one can see in the confusion matrix (Figure 14).

In conclusion, we still need a larger collection of data for each class—at least 1000 labels per class, because fewer than 1000 labels are not sufficient. Furthermore, our current dataset is disproportional for person labels; nonetheless, this class is useful for detecting whether a person is inside or not in the vehicle.

#### 5.1.4. Embedded System

Table 7 presents the inference times for aggressive objects using PyTorch models for different YOLO size layers (S = small, M = medium, L = large, X = extra large). The classes are knife, weapon, and bat. In terms of qualitative analysis, we determine that the Yolo_L has better precision results compared to Yolo_S and Yolo_M for aggressive object detection. The reason for why we don’t use Yolo_X is due to the amount of time to infer; it is almost twice as much as Yolo_L.

The 15 classes used for training were: Person, knife, weapon, bat, bag, book, phone, laptop, mouse, keyboard, bottle, cable, banana, apple, and orange. Table 8 presents the inference times for the object detector of lost items, using Yolo with a different number of layers. In terms of qualitative analysis, we follow the same strategy that is used for aggressive objects.

The following Table 9 shows the inference times (3 trials) using different levels of precision: single precision (32 bits), half precision (16 bits), and 8-bit integer of the network (Yolo_S, ‘yolov5s’) using TensorRT for Yolov5, with a pre-trained model with 80 labels.

In the quantitative results, as we can see from Table 9, that the inference time gap between FP32 and FP16 is, on average, 9 ms if we compare it with FP16 and INT8, and the difference is, on average, 4 ms. As such, it is almost half when compared to the 3 different levels of precision. When we qualitatively analyzed the results of the video in real-time, we concluded that the number of true predictions with INT8 is not as good as FP16, and compared to FP32 and FP16, the former has the same results for our use case. For these two reasons, the inference time and precision, we selected the FP16 precision level to use on our embedded system.

To validate our method, we used our test bench vehicle inside the laboratory. We also created an interface to show our results for the 3 methods. The following Figure 15 shows some of the results obtained from the algorithms, namely, lost items, aggressive items, and violent action detection, to validate our methodology.

## 6. Discussion

This article presents a system that is capable of detecting violent action behaviors within the scope of the SAV, and more specially, the implementation of algorithms for the detection of aggressive objects, lost items, and violent action. In the first stage, a search was carried out that was associated with the existing state-of-the-art algorithms suitable for performing the proposed tasks. The selected algorithms belong to the themes of object detection and action recognition. The first task was to detect violent actions. As a basis for training the selected algorithms in this component, it was necessary to create a dataset and to generate the respective labels. In this context, the internal MoLa InCar dataset was used. This task is performed not only by using RGB frames, but also when we fed the algorithms with thermal, NVS, and optical-flow features, which could be very useful in situations where we need light immunity.

After training and the respective evaluation of the results obtained, all of the models obtained good results; however, considering the balance between precision and real-time performance, our choice falls into the TSM (94.32%) architecture. For the object detection task, the strategy was similar for lost items and aggressive objects. In this case, we dont have an internal dataset to use for training, so the COCO and TAO were selected, which both were both filtered and manipulated with the MoLAnnotate toolkit to fit our format for development. For training, was used the YOLOv5 family of object detection algorithms was used to detect lost/forgotten items and aggressive objects inside the vehicle. Among the YOLOv5 models, the Small model is chosen for the task due to its similar performance metrics and lower complexity, as only 3 classes need to be detected for aggressive objects. Although we have more classes, namely 12 when we execute the lost items algorithm, we can still get obtained good metrics and real-time performance.

From the results obtained in Table 3, the NVS feature could be a very feasible data to iterate for iterating our future work, namely, by replacing the von Neunmann architecture to neuromorphic. Because NVS frames only capture information from the motions of the passengers, this could lead to a more efficient inference method in terms of processing and energy consumption.

## Figures and Tables

**Figure 1 sensors-23-05610-f001:**
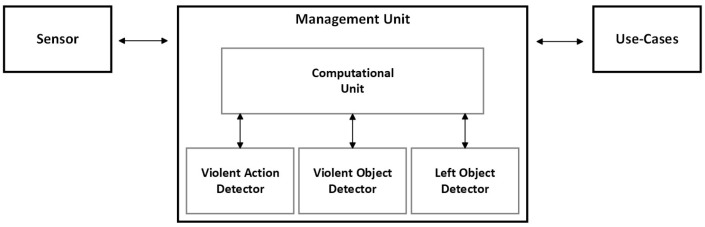
Architecture description of proposed solution.

**Figure 2 sensors-23-05610-f002:**
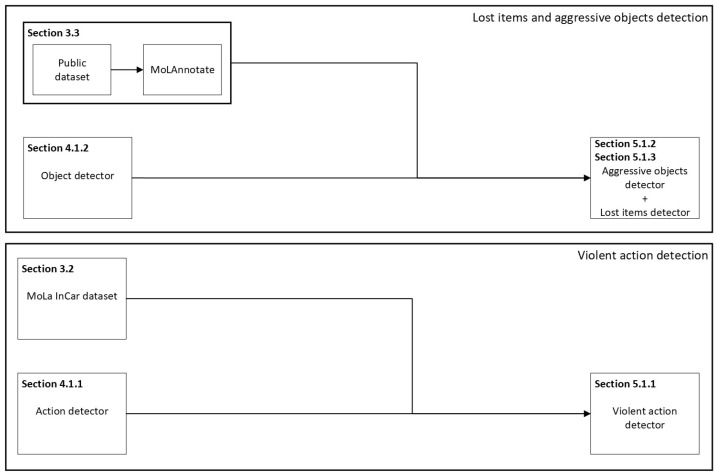
Two main development pipelines are presented: (1) lost items and aggressive objects; (2) violent action. Both with a focus on in-vehicle environment. For each pipeline, 3 identical steps are made: (1) a toolchain for data generation is implemented (Section 3.2 and Section 3.3); (2) models are selected (Section 4.1.1 and Section 4.1.2); (3) evaluate in (Section 5.1.1, Section 5.1.2 and Section 5.1.3).

**Figure 3 sensors-23-05610-f003:**
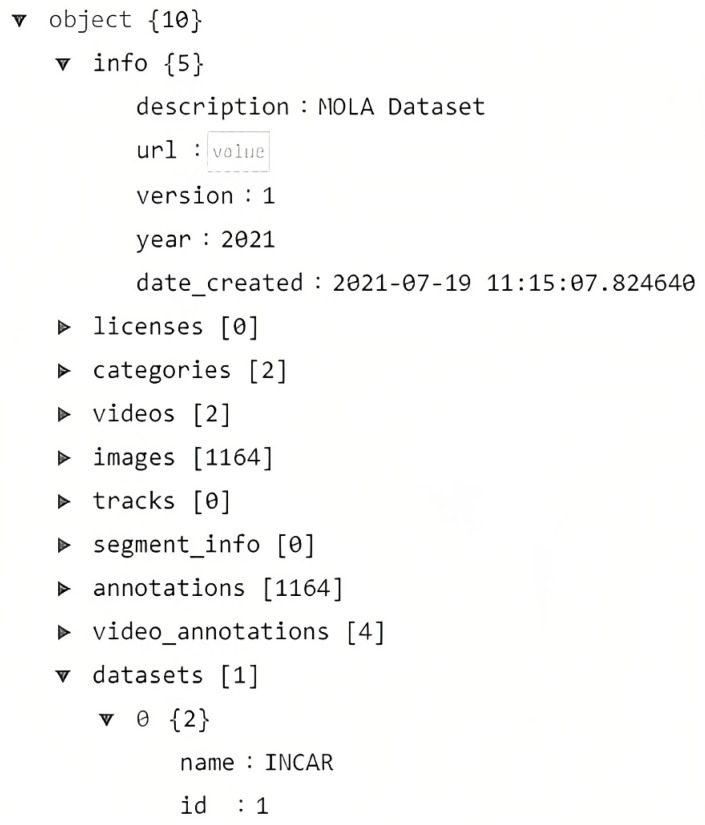
Annotation in MoLa data format.

**Figure 4 sensors-23-05610-f004:**
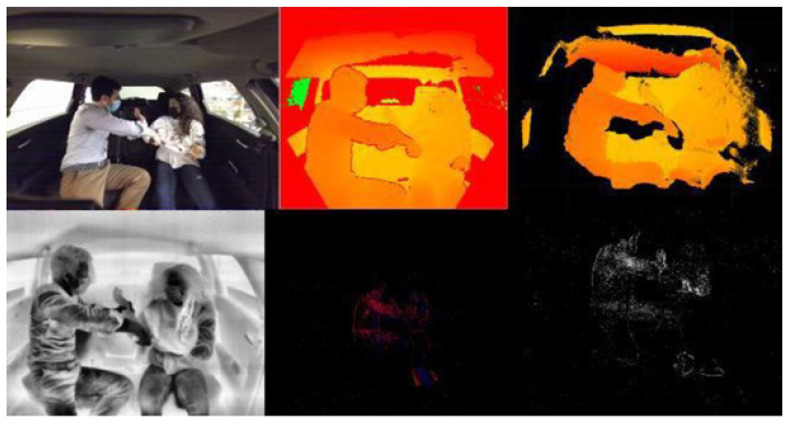
From Top Left to Bottom Right: RGB, Depth, Point-Cloud, Thermal, NVS, and Events Grayscale.

**Figure 5 sensors-23-05610-f005:**
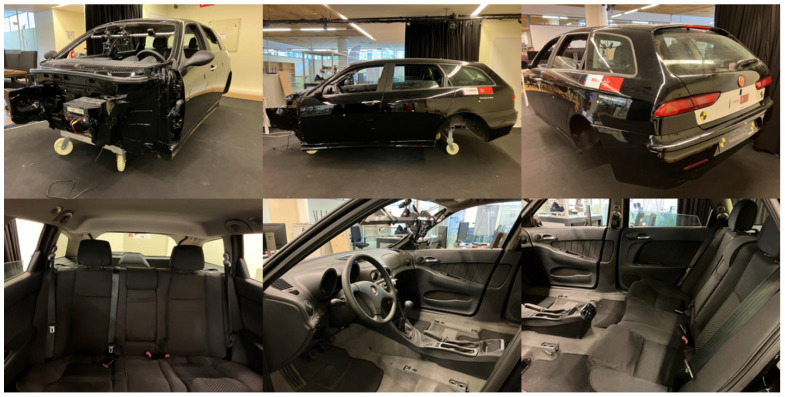
Laboratory car testbed from different perspectives.

**Figure 6 sensors-23-05610-f006:**
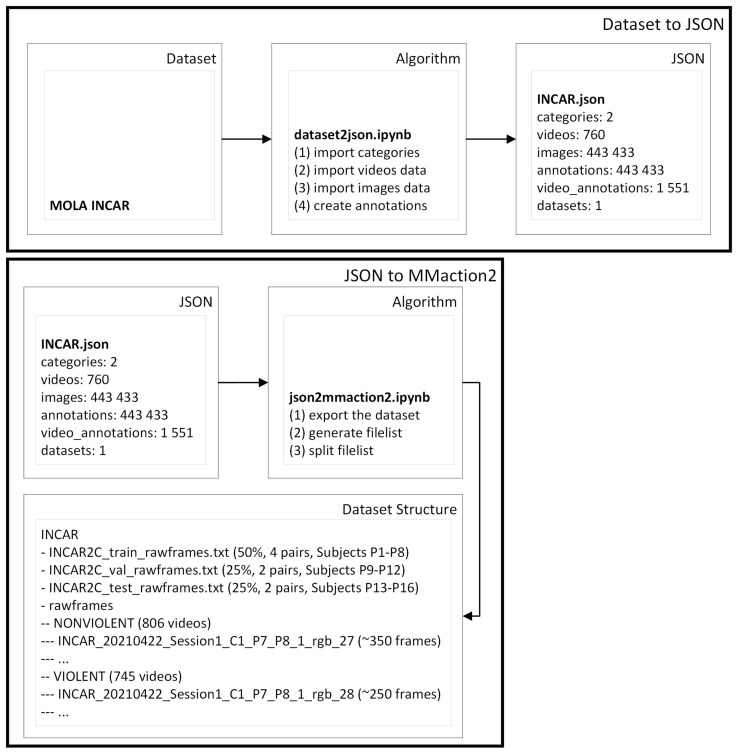
Dataset-to-JSON example.

**Figure 7 sensors-23-05610-f007:**
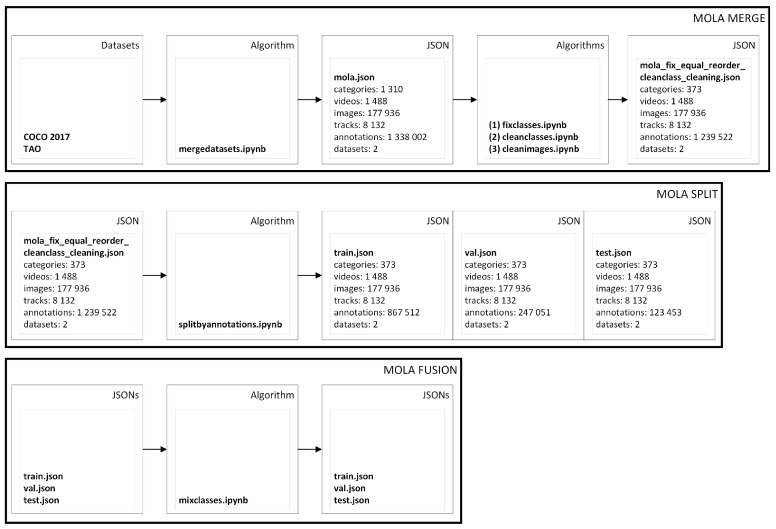
JSONs to MOLA example.

**Figure 8 sensors-23-05610-f008:**
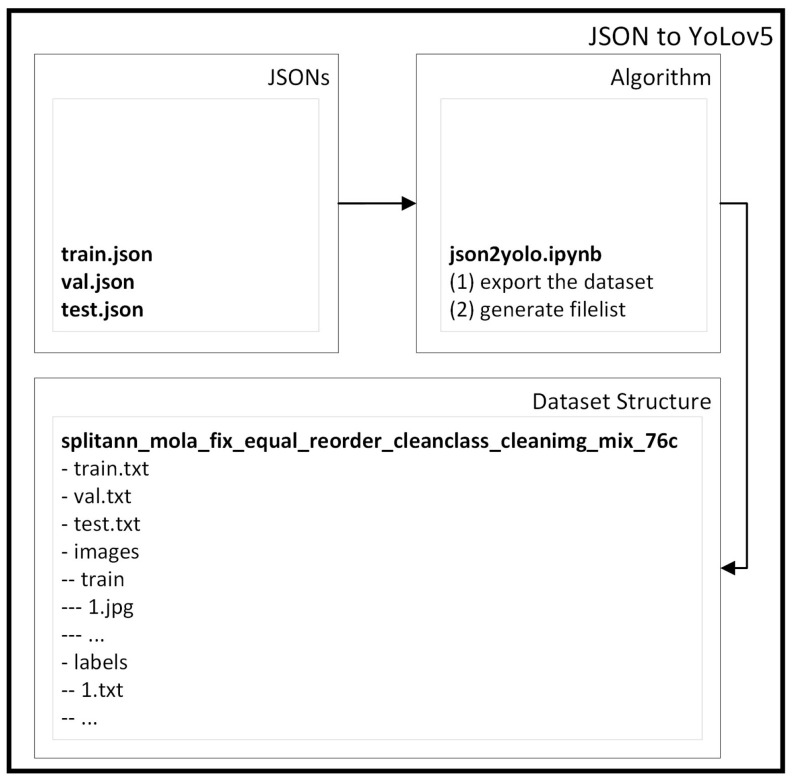
MOLA JSON to YOLOv5.

**Figure 9 sensors-23-05610-f009:**
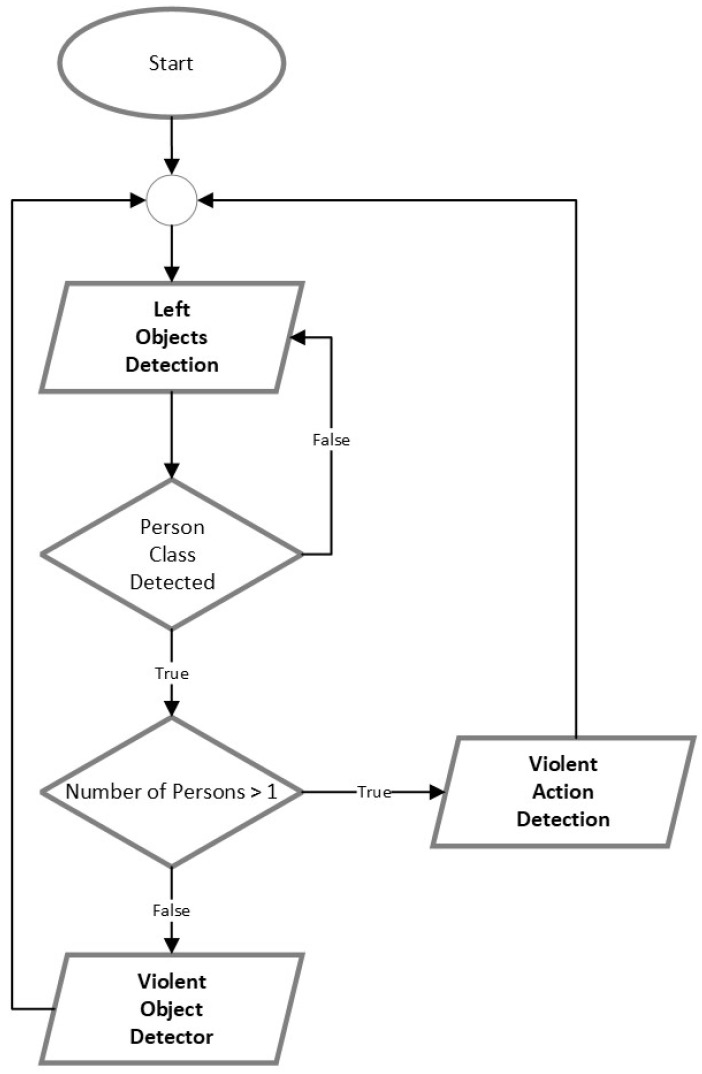
Pipeline for inference in NVIDIA AGX Xavier.

**Figure 10 sensors-23-05610-f010:**
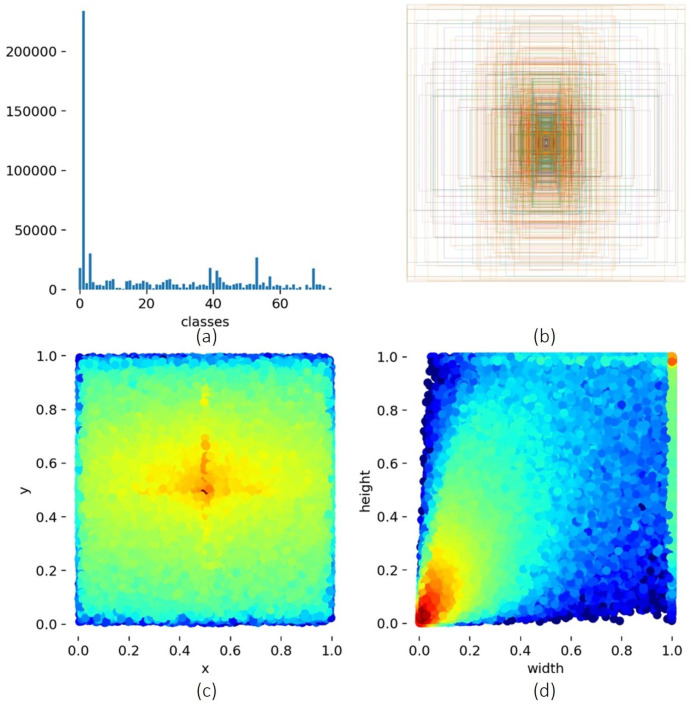
Visualization of 76 classes of the dataset. (**a**) Number of annotations per class. (**b**) Visualization of the location and size of each bounding box. (**c**) The statistical distribution of the bounding box position. (**d**) The statistical distributions of the bounding box sizes.

**Figure 11 sensors-23-05610-f011:**
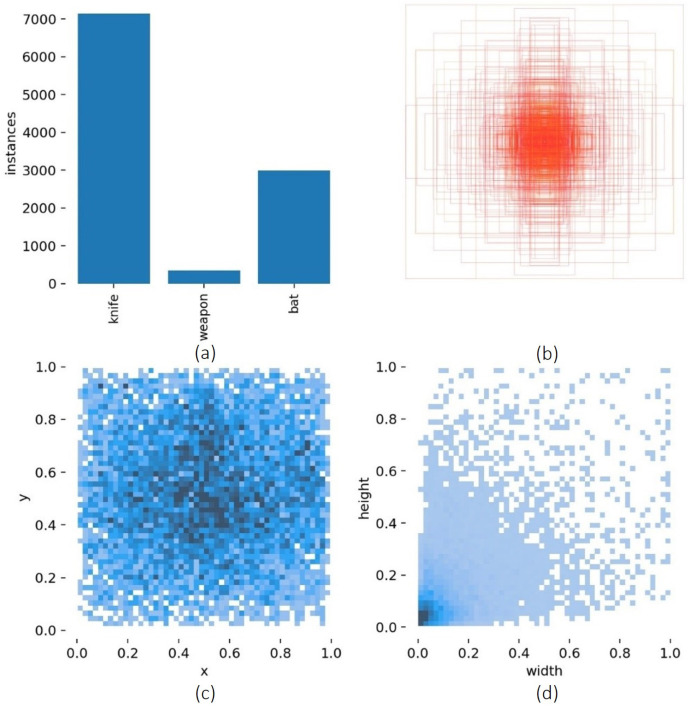
Visualization of the 3 most labeled classes of the dataset. (**a**) Number of annotations per class. (**b**) Visualization of the location and size of each bounding box. (**c**) The statistical distribution of the bounding box position. (**d**) The statistical distribution of the bounding box sizes.

**Figure 12 sensors-23-05610-f012:**
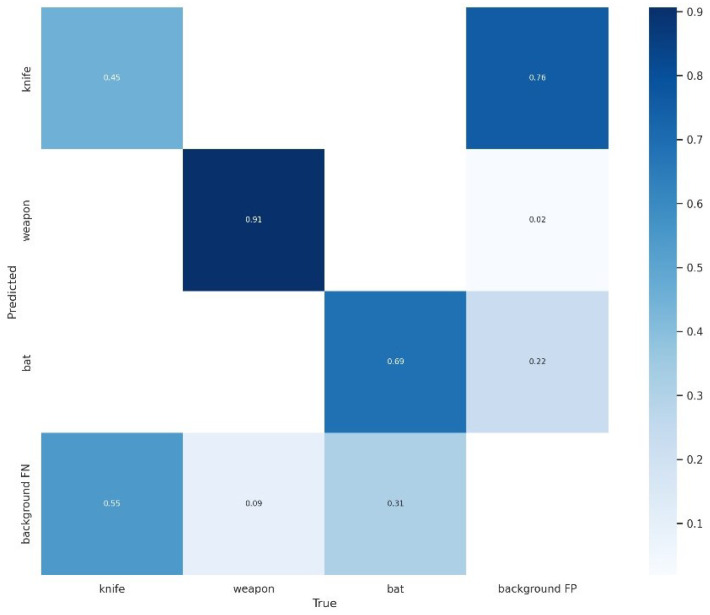
Confusion matrix of knife, weapon, and bat classes.

**Figure 13 sensors-23-05610-f013:**
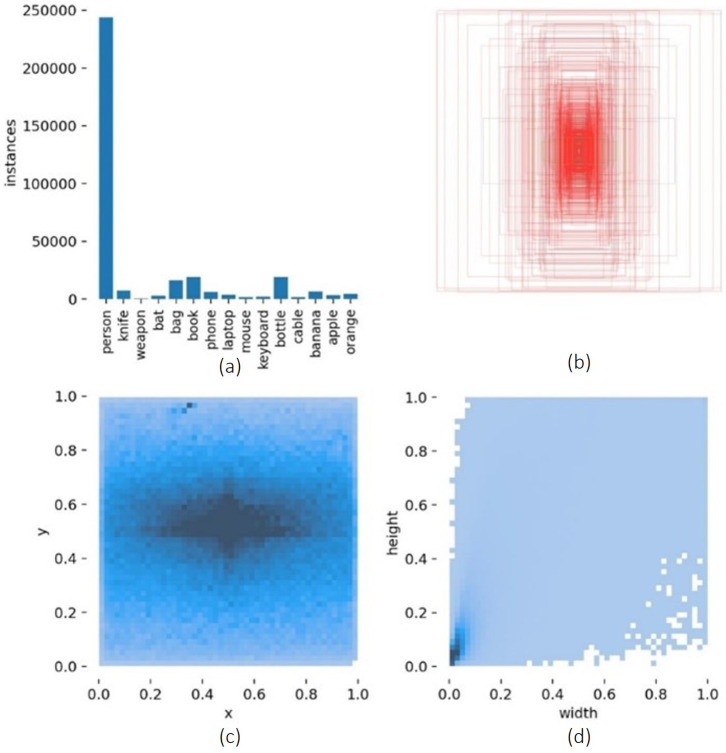
Visualization of classes for lost items objects on the dataset. (**a**) Number of annotations per class. (**b**) Visualization of the location and size of each bounding box. (**c**) The statistical distribution of the bounding box position. (**d**) The statistical distributions of the bounding box sizes.

**Figure 14 sensors-23-05610-f014:**
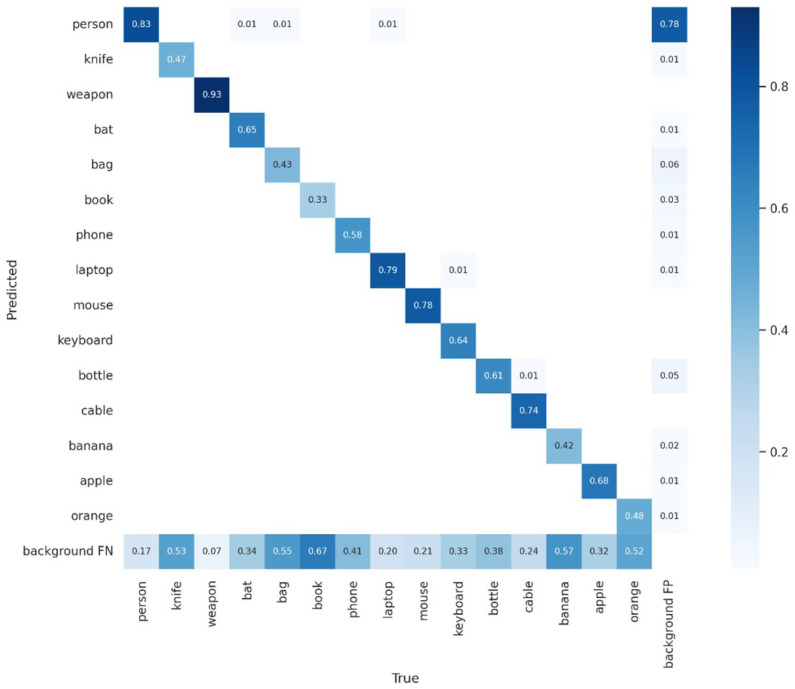
Confusion matrix for lost items.

**Figure 15 sensors-23-05610-f015:**
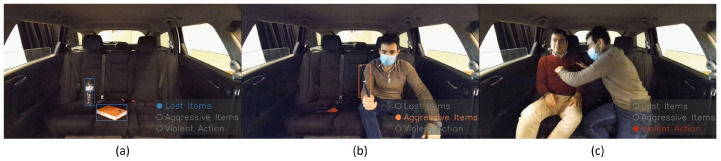
Examples of results obtained from real-time inferences. (**a**) Lost items detector. (**b**) Aggressive objects detector. (**c**) Violent action detection.

**Table 1 sensors-23-05610-t001:** Annotation Pipeline.

Process	Algorithm	Description
Merge	mergedatasets.ipynb	Merge different datasets
	fixclasses.ipynb	Find and fix duplicate classes
	cleanclasses.ipynb	Remove classes with missing annotations and images
	cleanimages.ipynb	Remove images missing from dataset folder
Fusion	mixclasses.ipynb	Mix/Fusion of classes into other classes & reorder class ids
Split	splitbyannotations.ipynb	Split using annotations in test, val and train
	splitbyimages.ipynb	Split using images in test, val and train
	reorderids.ipynb	Reorder class ids
Check	checkmissings.ipynb	Check missing images, videos and annotations
Export	json2yolo.ipynb	Exports dataset to yolo format
	json2mmaction2.ipynb	Exports dataset to mmaction2 format

**Table 2 sensors-23-05610-t002:** Inference accuracy for violent action.

Experiment	Methods	Top-1 Accuracy	Mean Class Accuracy
E1	I3D	0.7673	0.7737
E2	R(2+1)D	0.7737	0.7737
E3	SlowFast	0.5524	0.5546
E4	TSN	0.9540	0.9544
E5	TSM	0.9437	0.9432

**Table 3 sensors-23-05610-t003:** Best results per feature.

Feature	Experiment	Top-1 Accuracy	Mean Class Accuracy	Epoch
thermal	E4	0.9519	0.9452	20
thermal	E5	0.9500	0.9426	85
thermal	E5 (no pre-train)	0.8731	0.8402	75
thermal	E4 (no pre-train)	0.8846	0.8572	75
rgb	E4	0.9422	0.9263	5
rgb	E5	0.9827	0.9852	10
rgb	E5 (no pre-train)	0.9730	0.9788	40
rgb	E4 (no pre-train)	0.9634	0.9667	65
nvs	E4	0.8990	0.8891	35
nvs	E5	0.9732	0.9739	75
nvs	E5 (no pre-train)	0.8804	0.8721	75
nvs	E4 (no pre-train)	0.7216	0.6472	10
opticalflow	E4	0.9692	0.9633	50
opticalflow	E5	0.9885	0.9909	75
opticalflow	E5 (no pre-train)	0.8962	0.8730	65
opticalflow	E4 (no pre-train)	0.9077	0.8878	85

**Table 4 sensors-23-05610-t004:** YOLO Experiments.

Experiment	Precision	Recall	mAP_0.5	mAP_0.5:0.95
E1	0.126	0.225	0.0687	0.0331
E2	0.126	0.190	0.0597	0.0278
E3	0.143	0.262	0.0770	0.0375
E4	0.259	0.267	0.1410	0.0627
E5	0.546	0.662	0.5790	0.3810

**Table 5 sensors-23-05610-t005:** Test split metrics (1337 images).

Classes	Labels Test Split	Precision	Recall	mAP_0.5	mAP_0.5:0.95
all	1469	0.479	0.622	0.481	0.314
knife	1015	0.264	0.337	0.178	0.107
weapon	43	0.735	0.907	0.84	0.604
bat	411	0.439	0.623	0.426	0.232

**Table 6 sensors-23-05610-t006:** Metrics of the test split classes (38,048 images).

Classes	Labels Test Split	Precision	Recall	mAP_0.5	mAP_0.5:0.95
all	64,509	0.238	0.468	0.23	0.161
person	50,906	0.245	0.707	0.285	0.214
knife	1015	0.178	0.274	0.138	0.0929
weapon	43	0.734	0.9	0.773	0.525
bat	411	0.124	0.453	0.118	0.0725
bag	2332	0.0961	0.237	0.083	0.0549
book	2733	0.199	0.141	0.113	0.0714
phone	881	0.186	0.46	0.193	0.133
laptop	551	0.177	0.682	0.229	0.199
mouse	242	0.205	0.657	0.196	0.152
keyboard	306	0.215	0.497	0.186	0.139
bottle	2738	0.159	0.389	0.161	0.116
cable	243	0.616	0.646	0.617	0.388
banana	959	0.11	0.234	0.0773	0.0472
apple	491	0.191	0.483	0.184	0.134
orange	658	0.132	0.266	0.0959	0.071

**Table 7 sensors-23-05610-t007:** Inference times of aggressive object detection models using PyTorch.

Model	Num Classes	Inference Time (ms)
Yolo_S	3	120
Yolo_M	3	185
Yolo_L	3	300
Yolo_X	3	580

**Table 8 sensors-23-05610-t008:** Inference times of lost item detection models using PyTorch.

Model	Num Classes	Inference Time (ms)
Yolo_S	1298	175
Yolo_S	15	130
Yolo_M	15	270
Yolo_L	15	330

**Table 9 sensors-23-05610-t009:** Inference times for object detection using TensorRT.

Model	FP32 (s)	FP16 (s)	INT8 (s)
Yolo_S	0.024021	0.015262	0.010740
Yolo_S	0.023381	0.013463	0.010827
Yolo_S	0.022659	0.014394	0.012740

## Data Availability

The MoLa InCar dataset [6] was published and are available.

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
