# Peer review of "Fusion Object Detection and Action Recognition to Predict Violent Action"

_sensors, 2023, doi:10.3390/s23125610_

Round 1

Reviewer 1 Report

1. In abstract line 2,3,4 "This article focuses on the application of deep learning algorithms to detect objects, namely lost/forgotten items to inform the passengers, and aggressive items to monitoring if violent actions may arise between passengers". Rephrase the sentence to provide the actual meaning.

2.  Figure 3 "Algorithm" should be modified. The algorithm must be presented in table form not to be represented as figure.

3. Triangulation and comparison is the most important part of any scientific study. In the manuscript , I did not see these elements. Please try to compare it with others and make recommendation.

4. in Table 2 how the feature were selected?

5. Improve English quality. Some sentences are too long and some are to short that hide the actual meaning of the contents.

6. Recommendations and future direction of the study must be highlighted.

7. Why the proposed method and data set is used. The overall conclusion of the study is not mentioned.

8. Explanation of figure 10 must be provided in paragraphs.

9. Figure 11 and 12 should be represented in parts as 11 (a, b, c, and d).

Improve the quality and writing style. Make contents clear and concise to increase readability and effectiveness.

Reviewer 2 Report

This paper proposes an object detection method to monitor the in-vehicle environment of self-driving cars. The idea is intuitive and the experimental results also prove that the method is effective. However, several points should be revised and considered before acceptance. We note:

1. The introduction and related work section of the paper should be focused and divided into paragraphs that follow a logical relationship.

2. In line 70 of the paper, the citation numbers are not separated from the text. Please check this in the full text.

3. The proportion of papers published in the last three years in the references is too small.

4. The presentation of results and text labeling of Figure 16 are not clear enough.

There are some syntactic errors in the paper, such as "...scooter sharing, on-demand 20 ride services and ride sharing [3]." Please check the whole paper.

Reviewer 3 Report

This article focuses on the application of deep learning algorithms to detect objects, namely lost/forgotten items to inform the passengers, and aggressive items to monitoring if violent actions may arise between passengers.

1. In my opinion, starting the introduction with a Figure is not advisable.

Figure 1. Architecture description of purposed solution should be included in the methodology.

2. page 2 line 35 when it says "This type of method comprises two subclasses: traditional methods and deep learning 35

methods." references to literature are missing. Although the next chapter is related works.

3. page 3 line 79: Figure 1 or Figure 2??? it's not clear.

Figure 1. Architecture description of purposed solution

Figure 2. Architecture description of purposed solution. - the same beginning of the name of the drawing.

and in the description page 3 line 91: Figure 2 summarizes the entire development pipeline of this article.

Figure 2 is hard to read.

4. Workflow: Introduction. related works. - not conducive to formulating the problem. The purpose of the work is not clearly specified.

In Figure 1 or 2 where the proposed solution is, the existing solution and the new/proposed/innovative solution should be clearly marked.

5. Figure 3. Annotation MoLa data format. - very poor quality. I don't see the code.

6. Figure 9. MOLA JSON to YOLOv5 - very poor quality

7. Figure 10. Pipeline for inference in Nvidia AGX Xavier - I don't understand the diagram that there are 3 outputs from the conditional instruction (IF)? should be YES and NO or True - False. Aggressive object should exit after the loop/iteration ends.

8. What are the units of accuracy in Table 1 and Table 2? and in other tables?

9. There are no clearly formulated conclusions.

In conclusion, the research is interesting, but poorly presented, at a poor scientific level.

Round 2

Reviewer 1 Report

N/A

Reviewer 3 Report

The authors took into account the proposed amendments.

However, the quality of the Figures is still very poor.

The final chapter should be titled Conclusions rather than Discussion.